# Self-Sensing Properties of Fly Ash Geopolymer Doped with Carbon Black under Compression

**DOI:** 10.3390/ma14164350

**Published:** 2021-08-04

**Authors:** Cecílie Mizerová, Ivo Kusák, Libor Topolář, Pavel Schmid, Pavel Rovnaník

**Affiliations:** Faculty of Civil Engineering, Brno University of Technology, Veveří 95, 602 00 Brno, Czech Republic; mizerova.c@fce.vutbr.cz (C.M.); kusak.i@fce.vutbr.cz (I.K.); libor.topolar@vutbr.cz (L.T.); schmid.p@fce.vutbr.cz (P.S.)

**Keywords:** fly ash, geopolymer, carbon black, piezoresistivity, compressive loading, acoustic emission

## Abstract

The development of smart materials is a basic prerequisite for the development of new technologies enabling the continuous non-destructive diagnostic analysis of building structures. Within this framework, the piezoresistive behavior of fly ash geopolymer with added carbon black under compression was studied. Prepared cubic specimens were doped with 0.5, 1 and 2% carbon black and embedded with four copper electrodes. In order to obtain a complex characterization during compressive loading, the electrical resistivity, longitudinal strain and acoustic emission were recorded. The samples were tested in two modes: repeated loading under low compressive forces and continuous loading until failure. The results revealed piezoresistivity for all tested mixtures, but the best self-sensing properties were achieved with 0.5% of carbon black admixture. The complex analysis also showed that fly ash geopolymer undergoes permanent deformations and the addition of carbon black changes its character from quasi-brittle to rather ductile. The combination of electrical and acoustic methods enables the monitoring of materials far beyond the working range of a strain gauge.

## 1. Introduction

The production of concrete, the most widely used construction material, is responsible for approximately 8% of global anthropogenic greenhouse gas emissions and 3% of global energy demand, meaning that it has significant influence on the global environment [1]. Current efforts towards the mitigation of these impacts and a sustainable construction industry involve multiple strategies, one of which concerns the development of smart or multifunctional concrete that is intended to enrich a structural material by adding an extra function, and thereby improving durability, extending service life or reducing maintenance costs. The tailoring of such smart concrete to meet specific requirements is achieved through special composition design, processing or the introduction of functional components [2].

Concretes with enhanced electrical conductivity represent a broad subset of smart materials that are thus provided with various additional functionalities, e.g., self-sensing or self-heating behavior, electromagnetic shielding or energy harvesting [3].

Because the resistivity of ordinary concrete is high, generally in a range between 10^6^ and 10^9^ Ω·m [2], the application of a conductive filler is needed to maintain sufficient conductivity performance. The most common conductive fillers are carbon fibers, carbon nanotubes, graphite powder, carbon black, graphene and metal components. The self-sensing performance of concretes containing functional fillers is established by changes to the internal conductive network formed within the matrix caused by an external mechanical force or environmental action. Self-sensing composites are then characterized by increased capabilities to sense strain, cracks or damage via continuous resistivity measurements [4]. This concept of a structure being a piezoresistive sensor at the same time can lay the foundations for advanced solutions in the long-term structural health monitoring of civil structures, the weighing of vehicles in motion, traffic detection systems, etc. [5].

Although the majority of research studies concerning self-sensing materials are focused on traditional cement concrete, a minor number of investigations have dealt with polymer, asphalt or alternative binder matrices. Geopolymers have seen increased research interest in recent decades due to the wide range of suitable precursors available (natural, processed or waste materials) and their favorable mechanical properties, chemical resistance, durability and lower environmental impact [6]. Geopolymer synthesis is based on mixing pozzolanic aluminosilicate material (class F fly ash, metakaolin, burnt clays or others) with alkaline activator solution, which usually comprises alkali hydroxide and/or alkali silicate, preferably sodium- or potassium-based [7]. Based on life cycle assessment results, CO_2_ production per 1 m^3^ of alkaline activated/geopolymer binder greatly depends on source material selection, but these values are in any case significantly lower than that of benchmark conventional mortar made with pure Portland cement [8]. The nature of geopolymers combining the properties of inorganic binders and ceramic materials has shown potential for use in novel applications [9].

Regarding self-sensing abilities, geopolymers have been reported to possess higher electrical conductivity originating from their chemical composition, especially the increased availability of mobile hydrated Na/K ions supplied by alkaline activator solution [10] and Fe present in the source material [11]. The conductivity of geopolymers is generally influenced by the concentration of alkaline activator, binder/activator ratio and the frequency applied during resistivity measurements [12], yet the conductivity as such does not determine the quality of self-sensing performance [13]. Despite the studies concerning geopolymers without a functional admixture showing piezoresistive behavior, the application of conductive fillers is preferred in order to increase sensitivity and diminish the negative impact of fluctuating water content in the binder [13,14,15,16].

Carbon black is a colloidal form of elementary carbon, a common by-product of the incomplete combustion of organic compounds. It features a high specific surface, excellent chemical and thermal stability, and electrical conductivity [17]. The addition of carbon black reduces the electrical resistivity and enhanced the piezoresistive performance of cement-based composites [18,19]. In another study, carbon black was proved to be a convenient complementary admixture in combination with carbon fibers. The partial replacement of fibers (carbon fibers or carbon nanotubes) by particle filler results in improved workability while maintaining self-sensing potential [20,21]. A further advantage to the application of carbon black is the cost savings it brings, as it remains one of the most affordable conductive admixtures [17].

Acoustic emission (AE) is a non-destructive method with great potential that can be used for the real-time monitoring of a test specimen subjected to loading. AE is a passive technique which is used to detect changes in elastic energy which are caused by, for example, the external loading of a test specimen. An AE event, such as cracking, then propagates through the material in the form of an elastic stress wave towards the surface, where it is recorded by a piezoelectric sensor [22]. The advantages of AE include its easy and quick application, its relatively good resolution of cracking events and its ability to monitor in real time.

This paper reports the self-sensing properties of fly ash geopolymer mortars under different regimes of linear and cyclic compressive loading applied in experiments during which the performance of a plain material was compared with material modified by carbon black. The continuous measurement of a resistivity response under AC was accompanied by the monitoring of longitudinal strain and internal structural changes via an attached strain gauge and acoustic emission sensors.

## 2. Materials and Methods

Fly ash from the high temperature combustion of black coal (ČEZ, Dětmarovice, Czechia) was used as the basic cementitious material. Its specific surface area determined by the Blaine method was 340 m^2^·kg^−1^, and its chemical composition is given in Table 1. A sodium water glass solution (Vodní sklo a.s., Brno, Czechia) was used as the alkaline activator for the production of the basic geopolymer binder. The SiO_2_/Na_2_O ratio of the water glass was 1.6 and the density of the aqueous solution was 1.548 g·cm^−3^. Very fine Vulcan 7H carbon black (Cabot CS, Valašské Meziříčí, Czechia) was used as a conductivity enhancing admixture at dosages of 0.5, 1 and 2% with respect to the mass of the fly ash. In order to improve the dispersion of rather hydrophobic carbon black, an aqueous 2% solution of Triton X-100 dispersant (Sigma Aldrich, St. Louis, MO, USA) was used. Fine quartz sand with a grain size of up to 2.5 mm was used as aggregate to prepare the geopolymer mortars.

The composition of the geopolymer mixes is presented in Table 2. The preparation of the geopolymer mortars followed the given general procedure. At first, carbon black was suspended in a water glass solution with added Triton X-100. Then, fly ash and quartz sand were added gradually together with additional water to maintain sufficient workability and the mixture was stirred in a Hobart mixer for 5 min. The preparation of reference samples without conductive admixture followed the same procedure but without the addition of carbon black and Triton X-100. The fresh slurry was cast into 100 mm× 100 mm × 100 mm cubic molds and four electrodes made of copper mesh with a wire thickness of 1 mm were embedded in the specimens. The size of the electrodes was 80 mm × 120 mm, and the spacing between them was 20 mm. The experimental setup used for the measurements is depicted in Figure 1. The fresh specimens were at first kept for 2 h at ambient temperature and then cured for 24 h at 40 °C. The hardened specimens were wrapped in PE foil and stored under ambient conditions for 28 days. Then, the specimens were unwrapped and kept exposed to air in order to equilibrate moisture before testing.

A LabTest^®^ 6.250 (Labortech, Opava, Czechia), a precise electromechanical testing machine which enables a maximum working load of 250 kN, was used for the measurement of self-sensing properties in compression. The samples were fitted with a strain gauge to measure deformations in the direction parallel to the loading force.

The cubic specimens were loaded perpendicular to the plane of the electrodes in three different modes:Cyclic linear loading and releasing of the samples in the range of 0–10 kN with a loading rate of 300 N·s^−1^ and constant amplitude (Figure 1a).Cyclic linear loading and releasing with variable amplitude. The maximum force was 50 kN for the reference sample and 35 kN for samples with carbon black, due to their lower compressive strength. The loading rate was 500 N·s^−1^ (Figure 1b).Linear loading with a loading rate of 200 N·s^−1^ up to failure.

An Agilent 33220A sinusoidal signal generator and two Agilent 34410A multimeters (Agilent Technologies, Santa Clara, CA, USA) were used for the measurement of the electrical resistance of the loaded samples. The measurement was performed in AC mode with a frequency of 1 kHz in order to avoid polarization of the electrodes, and the input voltage was 5 V. The four-probe method was used; the experimental setup is presented in Figure 2. The tested specimens were electrically insulated from the press with 0.45 mm thick PE foil.

Piezoresistivity is described by means of fractional change in resistance (FCR), which is independent from loading conditions and the size of the specimen, and is commonly used to compare the self-sensing properties of different materials under various loading conditions. It is expressed as a relative change in resistance (Equation (1)).
(1)$$FCR\;(\%)=\frac{R-R_{0}}{R_{0}}\cdot 100$$

In order to express the sensitivity of piezoresistive properties to the strain response, a gauge factor (GF) can be used. It is defined as the fractional change in resistance per strain unit (Equation (2)), and helps to compare the self-sensing properties of different materials.
(2)$$GF=\frac{\Delta R/R_{0}}{\varepsilon }$$

Acoustic emission activity was generated as the material became damaged during compressive strength tests. The AE activity was monitored by a DAKEL XEDO multi-channel unit (ZD Rpety-Dakel, Rpety, Czechia) with the following input parameters: threshold value for individual AE hits 200 mV, sampling of AE hits at 4 MHz, and frequency range from 50–500 kHz. The total gain was 69 dB. The sensors were manufactured by DAKEL and have the model number IDK-09. The test also employed one sensor placed outside the test specimen on the press as a guard sensor. This guard sensor was used to filter out false signals from the environment during post-processing. The AE sensors were attached to the specimens with beeswax. The selected AE parameter for all evaluations was the root mean square (RMS). In the literature [23], it is highlighted that when failure is imminent, a considerable increase in the RMS can be noted. This is because the RMS is directly related to the signal energy.

## 3. Results and Discussion

### 3.1. Piezoresistivity and Acoustic Emission during Cyclic Loading

In the first experiment, the cubic specimens were loaded repeatedly with a constant force amplitude of 10 kN. Figure 3 shows changes in strain and electrical properties during compressive loading. The piezoresistive properties of a composite are based on changes in the conductive network inside it, meaning that changes in electrical resistance are able to characterize the deformations of the material that occur during mechanical loading. For these purposes, piezoresistivity is expressed as the fractional change in the electrical resistance of a material for different loading conditions. During compressive loading, the electrical resistance of the material decreased, which was caused by the healing of defects and the closing of microcracks caused by shrinkage. During relaxation, the resistance increased again.

The largest deformation during loading occurred in mixture CB 1, whereas the strain changes were similar for mixtures CB 05 and CB 2. The smallest deformation was then recorded for the reference mixture. This implies that the stiffness of the fly ash geopolymer decreases with the addition of carbon black. The gradual increase in the absolute strain after the compressive force is released shows that the permanent deformation of samples occurs due to the rather soft structure of the fly ash geopolymer.

In the case of a fractional change in electrical resistance, the lowest response to the load was observed for the reference material without the conductive admixture. During the test, a gradual increase in the electrical resistance of the unloaded sample was observed for all mixtures except for CB 2. The increase is more pronounced with increasing carbon black content. This phenomenon is related to permanent deformations, which are associated with the formation of defects and the opening of microcracks. On the other hand, the amount of carbon black in the CB 2 mixture almost reaches the percolation threshold. Therefore, the permanent deformation in this sample pushes carbon particles closer to each other and results in the achievement of the percolation of the conductive network, thus forming permanent conductive pathways that are responsible for the decrease in resistance.

The recorded AE activity represented by the cumulative RMS value is presented in Figure 4. The individual developments demonstrate that the highest activity was in the CB 05 mixture. Compared to the reference sample, this rise is related to the larger occurrence of microdamage due to weakening of bonding between particles in the internal structure of the material during the loading cycles, which resulted in decreased stiffness, but the material remained still quasi-brittle. An increased amount of carbon black (mixture CB 1) then resulted in higher ductility of the material, which was demonstrated by lower AE activity, which corresponds to a lower number of events related to damage. No AE activity was then recorded for the CB 2 mixture, though the permanent deformation is quite high. The signals were probably smaller than the preset sensitivity of the measuring setup. This implies that an increasing amount of carbon black results in a decrease in geopolymer stiffness and transforms the material from quasi-brittle to more ductile.

In the next experiment, the samples were repeatedly loaded with a variable amplitude (Figure 1b), which did not exceed 25% compressive strength at its maximum. Figure 5 shows the strain curves and fractional change in electrical resistance for all tested mixtures. From the viewpoint of deformations, it is evident that the strain rises with increasing content of carbon black. However, it must be taken into account that peak forces for the samples with carbon black are 30% lower than in the case of the reference sample. As with the previous experiment, permanent deformations can be observed and they are most pronounced for the CB 2 mixture (up to 730 µm/m).

The fractional change in resistance shows a relatively good response to the mechanical load, and the reproducibility of the measurement is also satisfactory. During loading, there was again an increase in the basic resistance of the unloaded REF, CB 05 and CB 1 samples, which is mainly associated with permanent deformations of the test specimens. This trend was most markedly observed with the reference geopolymer, while the increase was very small for the CB 05 mixture. The resistance of sample CB 2 decreased similarly to the previous measurement.

Corresponding gauge factors were calculated using numerical analysis and baseline correction of the FCR and strain data (Table 3). The data also show the initial resistance of the tested materials. Comparing the measured longitudinal strain and electrical response to the applied compressive load, the best self-sensing properties were obtained with mixture CB 05. The mean value of the gauge factor of this mixture was 6.0, which is only slightly lower compared to the results obtained for the cement composite with carbon fibers presented by Beaza et al. [24]. On the other hand, Chung [25] achieved a gauge factor for cement-based composites that was about two orders higher. The piezoresistive sensitivity of the reference geopolymer is very limited due to the absence of conductive admixture, though the initial electrical resistance is only slightly higher.

### 3.2. Piezoresistivity and Acoustic Emission under Monotonous Loading until Failure

In order to analyze the self-sensing properties throughout the whole range of loading force, the measurement was also performed during monotonous compressive loading until failure. The course of the loading was monitored in terms of the fractional change in electrical resistance, longitudinal strain and acoustic emission activity (Figure 6). The strain curves for samples REF, CB 05 and CB 1 are very similar. The longitudinal strain increased exponentially until the point when the stain gauge exceeded the measuring range or the sides of the cubic sample with the attached strain gauge delaminated (Figure 7). Mixture CB 2 exhibited a more linear and steeper increase; unfortunately, the strain gauge exceeded the measuring range well before the point of failure.

The fractional change in resistance curves related to mixtures REF, CB 05 and CB 1 exhibited a plateau or a slight maximum in the initial state of loading, followed by a steep decrease. Such behavior is a result of competition between the formation of preliminary defects which interrupt the conductive pathways and the healing of the microcracks caused by compressive forces. Since fly ash geopolymer is a rather soft material, there is no abrupt increase in FCR at the point of failure analogous to that which was reported for cement- or slag-based materials [11]. Mixture CB 2 showed an almost linear decrease in FCR, which is related to the increasing contact conductivity during compression. When the geopolymer matrix collapsed, FCR started increasing again due to the disintegration of the matrix and the interruption of conductive pathways (Figure 6d).

The graphs in Figure 6 simultaneously show the AE activity during loading until failure. In the case of the REF and CB 05 mixtures, the highest activity corresponds to the largest deformation of the test specimens. Due to its slightly lower stiffness but still relatively quasi-brittle behavior, mixture CB 05 shows much higher AE activity during almost the whole loading range when compared to the reference mixture. In the case of mixtures CB 1 and CB 2, the main AE activity is shifted more towards the moment when the matrix started collapsing, and it corresponds to the minimum in the FCR curve, especially in the case of the CB 2 mixture.

The dependence of the cumulative RMS on the longitudinal strain (Figure 8) clearly demonstrates the differences in the character of the individual mixtures. Since the data are only presented up to the maximum recorded compressive strain, we can only compare the initial part of the compressive test. The CB 05 mixture exhibited an almost linear increase in this dependence, which is related to the formation of a large number of microdefects, which, however, do not significantly affect the integrity of the material. The CB 1 mixture then demonstrates the effect of the added carbon black, where the AE activity at the beginning of loading is comparable to the reference mixture, but then increases significantly due to the lower compressive strength [26]. In the case of the CB 2 mixture, the curve is very flat because there is no significant AE activity in this part due to the higher ductility of the composite.

## 4. Conclusions

This work presents the results of testing the self-sensing properties of geopolymer composites based on high temperature fly ash with carbon black added as an electrically conductive functional filler. The piezoresistive response was studied during the cyclic loading of samples in compression in two modes, with constant or variable amplitude, and characterized by the measurement of the electrical resistance using an AC field with a frequency of 1 kHz. In the following experiment the specimens were subjected to monotonous loading until failure. The response of the material to the applied load was monitored by acoustic emission technique and by the measurement of deformations using a strain gauge.

Piezoresistivity was achieved for all tested mixtures, including the reference mixture without added carbon black. However, from the viewpoint of the calculated gauge factor, the best self-sensing properties were shown by the CB 05 mixture, which had 0.5% of carbon black admixture. The sensitivity of the reference geopolymer was very limited due to the absence of conductive admixture, and at higher dosages of carbon black the composite was getting closer to the percolation threshold, which again reduced its piezoresistivity.

The analysis of compressive strain and acoustic emission data also showed that fly ash geopolymer is not a sufficiently rigid material, and that during loading it undergoes permanent deformations which are associated partly with the formation of microdefects in the structure of the binder and partly with the change from quasi-brittle to rather ductile properties when carbon black is added. This was also reflected in the increase in electrical resistance of the unloaded material. The analysis using three different monitoring techniques showed that the simple measurement of strain does not provide sufficient information about the response of a material to mechanical loading. Strain gauge is affected by surface changes and represents the external effect of deformation on the loaded specimen. The combination of piezoresistivity and acoustic emission enables a deeper insight into a material’s behavior, and is able to provide self-monitoring far beyond the working range of a strain gauge because it monitors the effect of loading directly in the structure of the material and is not affected by surface changes.

The results of this study can be used in the design of new types of smart structures with self-sensing and self-monitoring properties. However, some aspects such as effect of internal moisture or shape and position of electrodes are yet to be investigated, and hence further research and experimental work is needed in this field. Future work will also focus on other types of conductive admixtures or their combination.

## Figures and Tables

**Figure 1 materials-14-04350-f001:**
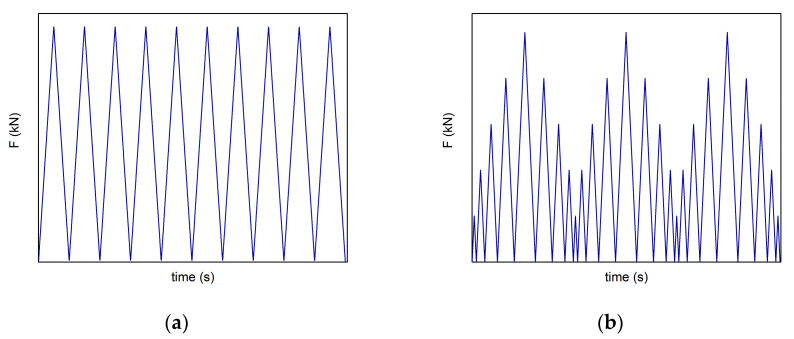
Loading regimes during a cyclic self-sensing test: (**a**) Measurement with constant force amplitude; (**b**) Measurement with variable force amplitude.

**Figure 2 materials-14-04350-f002:**
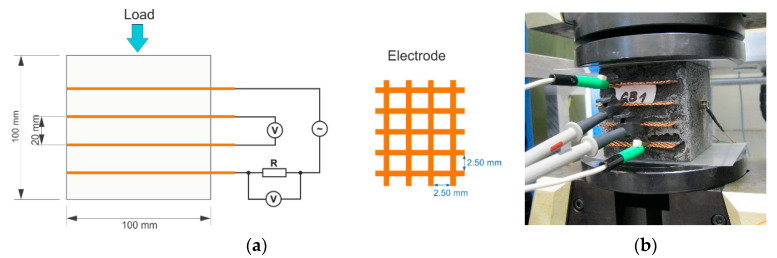
Experimental setup for the measurement of self-sensing properties under compression. (**a**) schematic plot; (**b**) picture of the setup during testing.

**Figure 3 materials-14-04350-f003:**
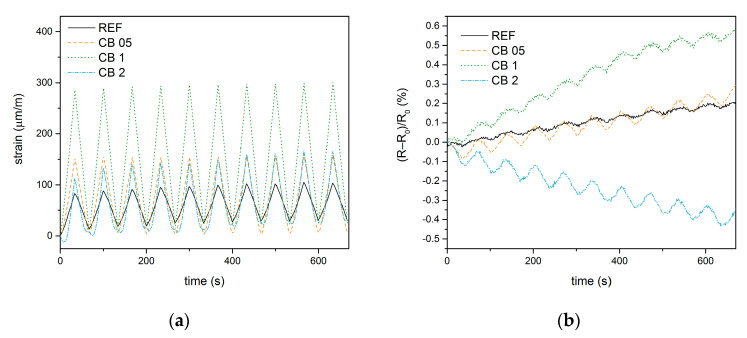
Self-sensing properties during compressive loading with constant amplitude: (**a**) Changes in longitudinal strain; (**b**) Changes in fractional change in resistivity.

**Figure 4 materials-14-04350-f004:**
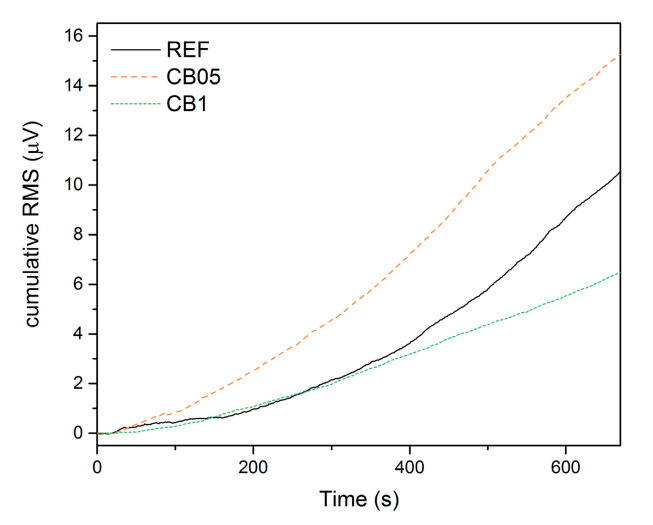
Comparison of cumulative RMS recorded for samples REF, CB05 and CB1 during repeated loading.

**Figure 5 materials-14-04350-f005:**
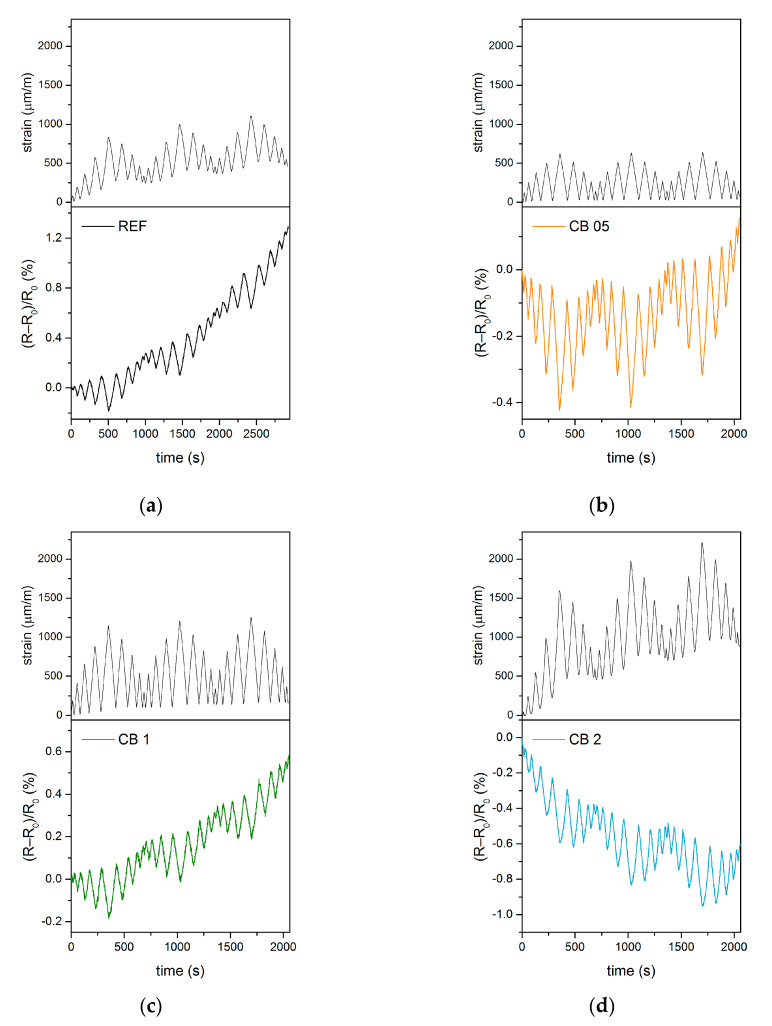
Self-sensing properties during compressive loading with variable amplitude: (**a**) Reference sample; (**b**) sample CB 05; (**c**) sample CB 1; (**d**) sample CB 2.

**Figure 6 materials-14-04350-f006:**
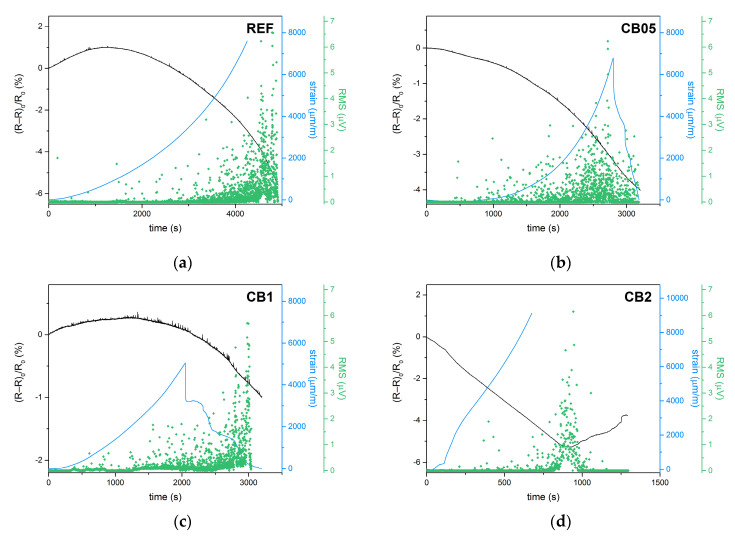
Self-sensing properties during monotonous compressive loading until failure: (**a**) Reference sample; (**b**) sample CB 05; (**c**) sample CB 1; (**d**) sample CB 2.

**Figure 7 materials-14-04350-f007:**
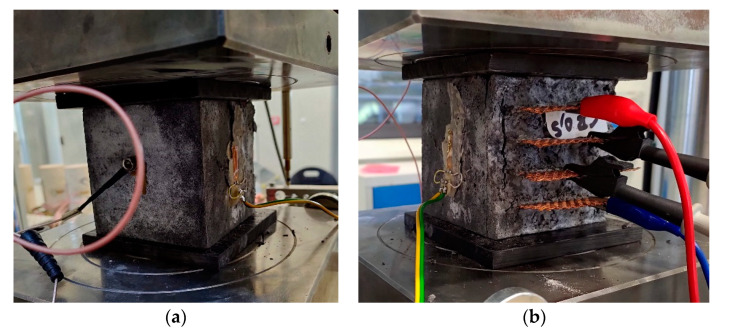
Delamination of the side wall of a specimen with an attached strain gauge. (**a**) back side; (**b**) front side.

**Figure 8 materials-14-04350-f008:**
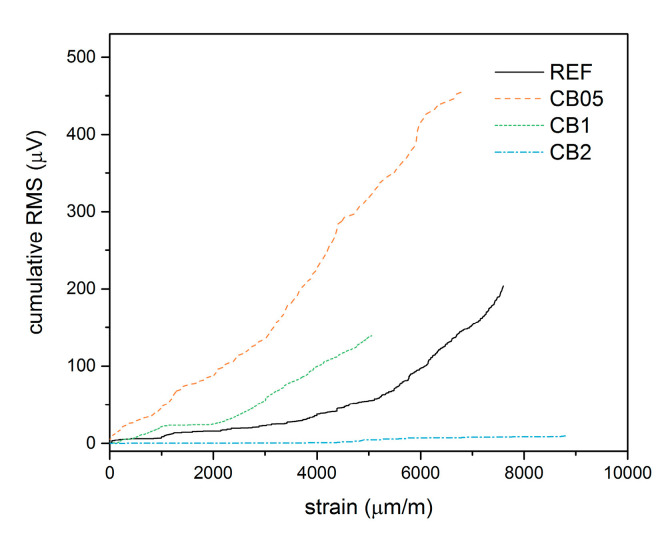
Comparison of cumulative RMS vs. longitudinal strain for all tested composite mixes. Data are only presented up to the maximum recorded compressive strain.

**Table 1 materials-14-04350-t001:** Chemical composition of fly ash (%).

SiO_2_	Al_2_O_3_	Fe_2_O_3_	CaO	MgO	K_2_O	Na_2_O	MnO	TiO_2_
51.67	23.31	7.08	4.45	0.36	2.95	0.77	1.14	1.00

**Table 2 materials-14-04350-t002:** Composition of the mixtures.

Component	REF	CB 05	CB 1	CB 2
Fly ash (g)	1200	1200	1200	1200
Water glass (g)	960	960	960	960
Quartz sand (g)	3600	3600	3600	3600
Carbon black (g)	-	6	12	24
Triton X-100 (g)	-	6	12	24
Water (g)	120	120	120	130

**Table 3 materials-14-04350-t003:** Calculated gauge factors.

Mixture	REF	CB 05	CB 1	CB 2
Gauge factor	2.53 ± 0.20	6.0 ± 0.08	1.75 ± 0.04	4.60 ± 0.33
Initial resistance (Ω)	36.8	33.6	22.1	24.3

## Data Availability

The data presented in this study are available on reasonable request from the corresponding author.

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
