# Peer review of "Self-Sensing Properties of Fly Ash Geopolymer Doped with Carbon Black under Compression"

_materials, 2021, doi:10.3390/ma14164350_

Round 1

Reviewer 1 Report

This is an interesting study on the self-sensing fly ash geopolymer. Please see attached pdf for review comments.

Author Response

Point 1: Why CB05 showed the highest activity? Then CB1 decreased comparing to the

REF, the explanation provided by the authors is not convincing.

Response 1: According to authors, this observation is quite clearly explained in the text. The addition of carbon black caused the weakening of the internal bonding between the particles in the geopolymer matrix. Since the CB05 mixture is still quasi-brittle, it displays higher AE activity. When the amount of carbon black is further increased, the structure transformed to more ductile, which resulted in a drop of AE activity.

Point 2: In conclusion, please provide discussion/suggestion for future study and potential

applications of this self-sensing geopolymer.

Response 2: A paragraph giving comment to future perspective of self-sensing geopolymers were added at the end of the conclusions.

Reviewer 2 Report

This work studies the piezoresistive response and the acoustic emissions in fly ash geopolymer specimens doped with carbon black under compression.

Certain comments and remarks towards the authors are highlighted below:

  • The specimens studied were selected to be of cubic shape with a side of 100mm. The authors have to explain why they chose this dimension and not a larger one. Would it have been better for the specimens to have a prismatic shape instead of a cubic shape?

  • The selection of mesh copper electrodes and their incorporation into the mixture which was poured into the molds, raises some questions about the reliability of the electrical resistance measurements. The copper must have oxidized during the curing process of the specimens. A very thin coating of non-oxidized metal on the copper electrodes should have been considered.

  1. No figures of the resistance change nor figures of the relative changes are presented, not even an indicative one. The percentage change of the resistance recorded in all tests is very small and does not exceed ± 2% until the specimens are severely damaged. Questions are raised about the reliability of the measurements.

  1. Regarding the AE recordings, in addition to the simple recording of Root Mean Square (RMS), authors should study other more important parameters, such as the acoustic activity expressed in AE hits per sec and the b-value analysis.

  1. The strain measurements during the monotonous compressive loading until failure, shown in fig. 6 for specimens REF and CB2, are not complete. The strain gauge was damaged well before the point of failure. These measurements should be repeated using other specimens and the presentation of the diagrams [fig.6 a and d] should be complete.

  1. The final conclusion reached by the authors [The analysis using three different monitoring techniques showed that the simple measurement of strain does not provide sufficient information about the response of a material to mechanical loading. The combination of piezoresistivity and acoustic emission enables a deeper insight into a material’s behavior, and is able to provide self-monitoring far beyond the working range of a strain gauge], is not convincing and is not sufficiently documented.

Author Response

Point 1: The specimens studied were selected to be of cubic shape with a side of 100mm. The authors have to explain why they chose this dimension and not a larger one. Would it have been better for the specimens to have a prismatic shape instead of a cubic shape?

Response 1: The size of the specimens was chosen due to limits of our equipment and testing machine. However, specimens of much smaller size (usually up to 40 mm) were used in most of the published work concerning self-sensing materials. The cubes were used because this type of specimens is commonly used for the compressive testing.

Point 2: The selection of mesh copper electrodes and their incorporation into the mixture which was poured into the molds, raises some questions about the reliability of the electrical resistance measurements. The copper must have oxidized during the curing process of the specimens. A very thin coating of non-oxidized metal on the copper electrodes should have been considered.

Response 2:  Yes, copper may have oxidized during the curing process. However, the self-sensing properties are commonly expressed in the way of fractional change in resistance/resistivity, which is independent from the absolute value of resistance.

Point 3: No figures of the resistance change nor figures of the relative changes are presented, not even an indicative one. The percentage change of the resistance recorded in all tests is very small and does not exceed ± 2% until the specimens are severely damaged. Questions are raised about the reliability of the measurements.

Response 3: Providing data for the resistance change would be redundant. Paper presents a relative changes in resistance (FCR) and initial resistances are given in Table 3. Simple resistance change would provide no additional information because the trends are the same. FCR is related to the longitudinal strain, which is expressed as gauge factor. The measured gauge factor is comparable with other published work (Ref. [24]), so should be no reason for misrepresentation of the measurement’s reliability.

Point 4: Regarding the AE recordings, in addition to the simple recording of Root Mean Square (RMS), authors should study other more important parameters, such as the acoustic activity expressed in AE hits per sec and the b-value analysis.

Response 4: The aim of the experiments was not to analyse the acoustic emission measurements in detail. It was primarily about the response of the material's internal structure to its stresses expressed in a simple form. The RMS level was chosen mainly because the threshold level was set too high, and therefore the acoustic activity in beats per second was minimal and did not show much difference. The RMS level was more sensitive in this respect and showed essentially the same thing, i.e. the amount of energy released by the acoustic signal.

Point 5: The strain measurements during the monotonous compressive loading until failure, shown in fig. 6 for specimens REF and CB2, are not complete. The strain gauge was damaged well before the point of failure. These measurements should be repeated using other specimens and the presentation of the diagrams [fig.6 a and d] should be complete.

Response 5: Unfortunately, it is not possible to repeat the measurement within such a short period of revision because all the specimens were damaged during measurement. The lack of data was caused by both exceeding the measuring range and the damage due to the delamination of sidewalls, which is presented in Figure 7. Anyway, though the strain gauges were not destroyed during the measurement of the other two samples (CB05 and CB1), they provided irrelevant data after the peak point also due to the delamination of side walls.

Point 6: The final conclusion reached by the authors [The analysis using three different monitoring techniques showed that the simple measurement of strain does not provide sufficient information about the response of a material to mechanical loading. The combination of piezoresistivity and acoustic emission enables a deeper insight into a material’s behavior, and is able to provide self-monitoring far beyond the working range of a strain gauge], is not convincing and is not sufficiently documented.

Response 6: Fig. 6 (a,d) shows where the capabilities of strain gauges end in monitoring individual samples during loading to destruction. Strain gauges only show the external effects of deformation on the specimen being loaded, while piezoresistivity and acoustic emission monitor the effects of loading within the structure. In particular, piezoresistivity measurements, where the grid itself is embedded directly in the structure under study, are not affected by surface changes.

Round 2

Reviewer 2 Report

I was expecting the authors to provide more concrete and to-the-point answers to several of my comments.

One of the main reasons I considered that the manuscript submitted is not suitable for publication was my reservations about the reliability of the measurements. I highlighted this point both in my comments to the authors and, also, to the editor.
I will only emphasize here “point 3” of my comments because I cannot devote additional time to the issue. According to my opinion scientific ethics requires that the directly measured quantities must be presented graphically and not just those resulting from calculations (such as percentage changes).
The authors ignored this comment arguing that it was not necessary. Moreover, the authors in the remark that: “the percentage change of the resistance recorded in all tests is very small and does not exceed ± 2% until the specimens are severely damaged”, answered by mentioning reference 24. However, this reference is based on the study of a different type of material and the resistance measurement is based on the “four-probe” method with non-embedded electrodes. In addition, in this work the measurements of the resistance are presented in a diagram followed by the % changes.
It is a pity that I have to apologize for my choice to suggest “Reject” of the manuscript.

Author Response

I have added a graphical output of originally measured data for the cyclic measurement together with appropriate comments to show that the trends in the resistance curves correspond to the trends in FCR curves. Since the changes in electrical resistance are dependent on the initial resistance, the fractional change in the electrical resistance was used to compare different mixtures. In order to avoid redundant data extending the paper enormously, further in the text, only FCR was used because, when initial resistance and percentage changes are provided, it is easy to imagine or calculate the real change in resistance. Although the changes in resistance are not very high, they provide a perceptible response to the applied load, and it is also in accordance with similar published work concerning cement-based mixtures.

Round 3

Reviewer 2 Report

I was expecting the authors to provide more concrete and to-the-point answers to several of my comments. 

The authors' response to the majority of my remarks are not convincing enough, and do not directly address the substance of the comments. 

I will only emphasize here “point 3”.  
According to my opinion scientific ethics requires that the directly measured quantities must be presented graphically and not just those resulting from calculations (such as percentage changes).

The authors ignored this comment arguing that it was not necessary. Moreover, the authors in the remark that: “the percentage change of the resistance recorded in all tests is very small and does not exceed ± 2% until the specimens are severely damaged”, answered by mentioning reference 24. However, this reference is based on the study of a different type of material and the resistance measurement is based on the “four-probe” method with non-embedded electrodes. In addition, in this work the measurements of the resistance are presented in a diagram followed by the % changes.